# Psychological and Social Vulnerability in Spaniards’ Quality of Life in the Face of COVID-19: Age and Gender Results

**DOI:** 10.3390/ijerph191610269

**Published:** 2022-08-18

**Authors:** Víctor-Raúl López-Ruiz, José Luis Alfaro-Navarro, Nuria Huete-Alcocer, Domingo Nevado-Peña

**Affiliations:** 1Department of Spanish and International Economics, Econometrics and History and Economic Institutions, University of Castilla-La Mancha, 02071 Albacete, Spain; 2Department of Political Economy and Public Finance, Economic and Business Statistics and Economic Policy, University of Castilla-La Mancha, 02071 Albacete, Spain; 3Department Business Administration, University of Castilla-La Mancha, 13071 Albacete, Spain

**Keywords:** age, gender, COVID-19, vulnerability, quality of life

## Abstract

Following the serious health situation in Spain and around the world in 2020 and 2021 stemming from COVID-19, this paper examines how the most vulnerable groups (in social and health terms) in Spanish society suffered the worst impacts on their quality of life during the pandemic. The review of the literature and publications released by the health authorities highlight the high number of cases of illness and death due to COVID-19; however, there are no studies about how it has affected aspects of citizens’ daily lives, considering some of their sociodemographic characteristics such as age and gender. A key issue when measuring citizens’ quality of life is that we typically see a U-shaped curve by age group, where young people register the highest shares along with the elderly; nevertheless, the pandemic has clearly brought about a change in this regard. In this study, we corroborate this finding, along with the psychological issues underlying this fact in young people and the physical ones in the elderly. To do so, we use an ANOVA and regression analysis with data from a sample of 908 Spanish citizens belonging to different age groups. These data were collected through an online survey distributed throughout Spain, mostly by email and social media, between 15 February and 15 April 2021 with a margin of error of 3.25% and a confidence level of 95%. Furthermore, the analysis allowed us to determine how aspects of daily life—family situation, living conditions, social environment, employment and financial situation—have been negatively affected during the pandemic, clearly depending on the age of the people surveyed, pushing them towards social and health vulnerability.

## 1. Introduction

COVID-19 has had significant social and health consequences worldwide, negatively affecting economies, and especially health sectors and educational systems [1]; however, its adverse effects have also been felt by certain more vulnerable social groups [2]. These consequences can be extensive and long-lasting, not only affecting the most vulnerable but also leaving consequences for future generations [3].

Vulnerability affects individuals differently depending on their social disadvantages. For example, social vulnerability refers to the strength of communities when faced with complex external factors such as the social rupture caused by COVID-19 [1]. This phenomenon has posed important challenges for society, causing increases in depressive and obsessive-compulsive symptoms [4]. Social distancing measures have had extremely negative effects worldwide, giving rise to social instability mainly among vulnerable populations; for example, in low-income households with children [5], the well-being of families is of particular concern due to the potential lasting effects on health and cognitive and social development. However, for both citizens of working age [6] and those still participating in some stage of education, age and vulnerability with respect to COVID-19 are key determinants of their stress levels, coping and self-esteem, shaping future social structures in the medium and long term.

Thus, the COVID-19 pandemic has changed the day-to-day lives of all citizens. More specifically, taking into account people’s ages, it has a priori had a more immediate impact on the lives of older adults because they are more likely to suffer serious illness or even death caused by the coronavirus, and they also faced greater social isolation during the pandemic [7,8]. The greater vulnerability of this older age group to the pandemic has prompted a substantial stream of research examining this impact and their well-being [9]. On the other hand, thanks to the success of public health policies and socioeconomic development, we have achieved longer life expectancy. This has brought with it an increasingly ageing population, with high rates of fragility and vulnerability, for which we do not always have appropriate quality-of-life indices [10] that, ultimately, also lead to obtaining happiness, if one takes into account their place of residence, public goods and the services they provide [11]. In this sense, it has been shown that happiness is linked to quality of life since it is one of the key factors in subjective well-being and general satisfaction with life [12,13,14].

The rise in the number of elderly people due to the ageing of populations is a challenge that societies must face. There is a wide range of vulnerabilities that can occur in old age, such as health-related loss of consciousness, specific physical needs, abuse, or financial limitations. They all have very significant influences on the quality of life of older people [15]. However, there are other dimensions of vulnerability that can emerge over the course of one’s life, such as long-term unemployment, employment precarity, and eventually retirement [16]. In many countries, retirees do not receive adequate financial support, despite the fact that their health expenses are usually higher. This makes older people vulnerable to economic insecurity [15].

Thus, the diverse nature of human beings and their heterogeneous responses to ageing have been the subject of extensive academic research [17]. Ageing itself is associated with natural physiological, structural and functional changes; however, in some people, ageing occurs at a much more accelerated rate, while others remain fit and healthy even at a much more advanced age. Pre-pandemic data indicated that people over 60 represented 12% of the world’s population, with this figure forecasted to rise [10]. However, the emergence of COVID-19 has tended to position older adults as an at-risk population, making them the most vulnerable [8,18]. Despite this, it seems that older adults generally cope better with this exceptional situation than younger ones. Possible explanations for this may include the more uncertain working conditions and therefore more serious financial problems that young people have been suffering over the last decade [19]. However, other studies such as that by Almeida and Santos [20] posit that the closure of companies due to the pandemic may make older workers the most vulnerable group, since they face difficulties in finding a new job but are still too young to take retirement [20].

Looking now at the group of young people in more depth, it is worth first providing a definition. Following INJUVE [21], we define young people as citizens aged between 15 and 29 years old [22,23], a value centred between those given by certain international institutions, which define youth as the stage of life up to 25 years old (UN), and studies such as that by Pieh et al. [19], which extend the period up to the age of 35. Having defined the age group, the challenges they face due to COVID-19 can be seen as additional to the risks identified as affecting this vulnerable group; for example, their financial emancipation, which is dependent on their entry into the labour market. In terms of the consequences of this vulnerability among the youngest citizens, the impact has tended to fall on the education they receive or consequences for their health, such as their eyesight after so much use of electronic media [3,24,25]. However, some authors such as Drane et al. [1] argue that, despite this, it is important not to problematize the individual or assume that this is a group that requires assistance. That said, the results of their study contradict this statement because the outbreak of the pandemic and the closure of classrooms at all levels meant that the most vulnerable young students suffered disruptions to their education. The onset of the pandemic forced many education systems to use learning alternatives such as remote electronic learning, which had consequences for behaviour and health, especially for the youngest [3,24,25]. In this regard, other researchers such as Perry and McConney [26] and Masten and Motti-Stefanidi [27] have claimed that students in communities of low socioeconomic status are either unfamiliar with cultural or social capital in conventional educational settings or tend to experience lower levels of community interconnectedness and shared resources. This aggravating factor has caused a high level of instability for the youngest citizens, as they require a certain degree of constancy in the midst of the urgent changes made to help them process and define new strategies to deal with emerging situations, such as the COVID-19 pandemic [1].

If we focus on the young population in education, it seems to be one of the hardest-hit groups after these two years of the pandemic, for reasons related to anxiety and stress [28], with negative long-term consequences especially for their mental health. Different researchers focusing on various societies, such as Pieh et al. [19], Ahmad et al. [29], Rossi et al. [30], Silva Moreira et al. [31] and Ueda et al. [32], examined whether the influence of COVID-19 on mental health depends on some of the individual’s sociodemographic characteristics. Ahmad et al. [29] concluded that the pandemic in India caused 25% of the population to suffer from anxiety, which was correlated with sex, income, employment status and marital status. Rossi et al. [30], focusing on Italy, showed evidence of a high percentage of post-traumatic stress symptoms, specifically, 37% of the population. Age and gender were found to be notable elements of this profile, with young women being the most vulnerable in terms of mental health. Also in agreement on this, the findings of Pieh et al. [19], in concert for the case of Austria during the time of COVID-19, finding the most vulnerable to be young people under the age of 35, mostly women, people with low incomes and people out of work. However, it is interesting to note that these results are the inverse of those from pre-COVID-19 studies in Austria, where mental health was found to be decreasing with age, by almost 78% in the age range between 15 and 30 years and by 70% in the population over 75 years of age [19]. In Japan, vulnerability to the pandemic also predominantly affects the youngest and the unemployed [32], in line with the conclusions from the study by Wang et al. [33] for China. Lastly, Silva Moreira et al. [31] focusing on Portugal, concluded that there has been an increase in depressive and obsessive-compulsive symptoms during the pandemic but that having a job protects against these symptoms.

Looking further at this social group, we can add other analysed perspectives. Dosil-Santamaria et al. [28] found that in psychological terms, the most vulnerable group of young people in Mexico were university students. These authors studied the mental health of young people after 18 months of the pandemic and explored how sociodemographic variables can influence symptoms. In their study, more than half of the sample (60%) had been in contact with a person affected by the virus; almost 30% had been ill with COVID-19; and almost 40% had lost a loved one. One particularly notable finding from this study was that 81% of respondents confirmed that they were not complying with the rules established to prevent the spread of the virus. On the other hand, in contrast to previous research, Dosil-Santamaria et al. [28] did not find significant differences between university students under and over 21 years of age [19]. “Young people who suffer from a chronic illness or who are close to someone who is ill tend to suffer psychological consequences [28]”. Consequently, even compared with pre-COVID times, the stress level is much higher among younger students. The trend has been confirmed in previous studies [34], where the average stress score is 12.7 in adults aged 20 to 39 years but reaches 19.3 for young people aged between 18–24 years, followed by 17.9 (25–34 years) and 16.2 (35–44 years).

Other researchers such as Dale et al. [35] showed that while the younger age group (18–24) was the most distressed and showed significantly more mental health symptoms compared with the older age group (65+) related to depression (50% vs. 12%), anxiety (35% vs. 10%) and insomnia (25% vs. 11%, all *p*-values < 0.05). Mental health has improved compared with the first lockdown in early 2020. Further analyses indicate that these findings were especially evident for those under 24 years of age, females, single/separated, low-income, and those who did not participate in any physical activity (all *p*-values < 0.05). We highlight the need for ongoing mental health support, particularly for the most burdened groups.

On the other hand, other research concluded that an action plan must be implemented to address the psychosocial and mental health needs of vulnerable children and adolescents during and after the pandemic, such as providing access to support services [36].

Furthermore, within the youth population, if we consider gender, the findings indicating that women have suffered more than men have appeared to be significant. Those who worked were more affected, given the difficulties in achieving a work–life balance during lockdown [37], with women more often having to take care of other family members than men. The gender differences that already existed in pre-pandemic times have thus been accentuated [38].

Finally, COVID-19 did not only affect the education of the youngest in society; according to Almeida and Santos [20], the most vulnerable group was people of working age who were largely unqualified, especially those with precarious job contracts. They claimed that highly qualified young people will have new opportunities to work due to the incorporation of new digitalization processes [20].

Against this backdrop, the aim of this research was to analyze the most vulnerable groups established by gender and by age range, accounting for the influence of COVID-19 on certain factors influencing Spanish citizens’ quality of life: namely, living conditions, family situation, employment status, income level, well-being and health. To this end, we conducted a review of vulnerability in citizens based on the determinants of their quality of life and how the pandemic had affected them depending on the sociodemographic characteristics of age and gender. We here describe the data collection and the characteristics of the sample. By conducting an analysis of variance (ANOVA) using different age groups and a regression analysis considering only two age groups, we obtained clear results on which to base discussions about the differences in the impacts on citizens’ quality of life linked to gender and age. We can thus draw conclusions on the consequences of the pandemic for Spanish society in terms of vulnerability.

## 2. Materials and Methods

### 2.1. Measures and Instruments

First, the data were collected through an online survey distributed throughout Spain, mostly by email and social media. The survey was conducted during the pandemic, specifically between 15 February and 15 April 2021. The resulting sample had 908 complete answers. The survey had a sampling error of 3.25% with a confidence level of 95%.

The questionnaire was divided into three sections. The first section gathered sociodemographic data on Spanish citizens, such as age, gender, marital status, place of residence, and job sector. In this case, age and gender are the characteristics we will use to differentiate between respondents’ vulnerability. The second section captured variables related to the measurement of citizens’ quality of life and happiness. On the one hand, the survey included a direct assessment through items related to key aspects of quality of life and the respondents’ surroundings, such as family situation; trust in the neighbourhood; trust in the foreign-born population; care for the environment; commercial accessibility and public transport; cultural and sports facilities and activities in their place of residence; available health and educational services; housing prices and safety of the residential environment. These key aspects of citizens’ quality of life were developed based on a literature review, specifically, Florida et al. [11]; Diener et al. [12], Diener [13] and Moghnie and Kazarian [14]. On the other hand, it included items related to respondents’ educational and employment situation, that is, employment status, education received, working environment, remote work and financial situation. The third section was aimed at collecting information about how COVID-19 had affected citizens. To this end, there were two questions, one directly asking about the effect on their quality of life and the other asking about which aspects of their lives had been the most affected by the pandemic. Specifically, this last section focused on the respondents’ surroundings, distinguishing aspects such as personal and family situation; employment and economic situation; training and education; place of residence; or not having been affected by the pandemic.

All the questions included in the first two sections were answered on a 1–10 Likert-type scale, where 1 indicated “not at all satisfied”/“disagree” and 10 indicated “very satisfied”/“agree”. Of all that information, Table 1 shows the variables prioritized and used in this study.

It is worth clarifying that QOL01 and QOL14 measure, respectively, respondents’ degree of happiness a priori and a posteriori, that is, after taking into consideration the different aspects analysed in the survey. The variable QOL13 analyses the effect of COVID-19 on respondents’ quality of life and the rest of the variables capture different aspects that affect quality of life, considered from a multidimensional point of view. Thus, with the information collected, a first descriptive study of the sample was carried out before going on to perform an ANOVA and then a regression analysis. The ANOVA was used to analyse whether there are differences in the means of the different variables listed in Table 1 depending on the age and gender of the respondent. To this end, different age groups were considered: 18 and 19 years old; between 20 and 29 years old; 30 to 39 years old; 40 to 49 years old; 50 to 59 years old; 60 to 69 years old; and over 70. For the regression analysis, two age groups were established in order to be able to fit two models and compare the results: young people (29 and younger) and older (over 29), according to the INJUVE criterion [21] and other authors such as Singal et al. [22] and Mehrdadi et al. [23]. These analyses were carried out using the statistical environment R [39] in the case of ANOVA and 12 Eviews for the regression analysis.

### 2.2. Study Sample Characteristics

Table 2 below shows the distribution of the sample of 908 Spanish citizens by age range, where young people (up to 29 years old) represent 24.2% of the sample. Of the rest the sample, the group of citizens aged between 40 and 49 stands out, with a percentage of 30.7%. Citizens aged between 18 and 19 years old and those over 70 were the least represented in the sample. Furthermore, the percentage of women participating in the survey (61.3%) is greater that of men (38.7%), and those who are married or living as a couple (65.1%) are more represented. Regarding the place of residence, almost half of the sample (44%) lived in municipalities with between 1000 and 5000 inhabitants. They were also asked about their employment status and job sector. The results revealed a high percentage of unemployed workers during this pandemic period (15 February to 15 April 2021), with the country’s economy stagnating.

Furthermore, the responses given by Spanish citizens about how the pandemic had affected them mostly (more than 90%) indicated a negative impact on quality of life. The mean score for this answer also provides the first indications related to this question, with the negative influence on quality of life reaching 7.96 points on a scale of 1 to 10. However, the results showed that people aged between 20 and 29 had a high mean score (M = 8.45; SD = 1.87), with those over 70 registering an even higher mean (M = 8.53; SD = 1.64). As such, it was the results from the direct answer that indicated the greatest impact from the pandemic. The analysis also confirmed how the COVID-19 pandemic affected citizens’ surroundings and where this impact was most serious. For example, at the personal/family level, the effects were noted by 63% and by 18.39% at the work level, followed by the educational sphere with 10.46%. In addition, the study shows that only 6% of the population reported a “denial” of the disease, stating that it had not affected their quality of life.

## 3. Results

First, we carried out an ANOVA, which allowed us to determine whether there were significant differences in the means registered by each of the age groups for the different factors that influence Spanish citizens’ quality of life. The first decision to make was which statistic we should use to carry out the ANOVA based on whether or not there was homogeneity of variances. To this end, Table 3 shows the results of Levene’s test for the homogeneity of variances taking gender and age group as factors. In the cases where the variances were not homogeneous, we used the Welch statistic (W), while when the variances were homogeneous, we used the F statistic (F).

In light of these results, we ran the ANOVA to determine whether there were significant differences in the means registered for each of the variables between the groups established on the basis of the two factors. The main results are presented in Table 4.

The results in Table 4 indicate that when accounting for the gender of the respondents, there are statistically significant differences only for the variable QOL05: men have a higher mean score for their level of satisfaction with access to public transportation and shopping areas (M = 7.3; SD = 1.99) than women (M = 6.99; SD = 2.20). This situation, together with question QOL12 with a value close to significance at 90%, seem to point to issues related to the workplace gender gap, due to evaluations of the working environment, the circumstances related to mobility and the management of time dedicated to shopping. Therefore, it can be concluded that there is no difference at 5% significance in the means assigned to the different variables that affect quality of life or in their evaluation when comparing the groups of men and women except for QOL05. Moreover, there are no significant differences in the effect of COVID-19 on quality of life. Table A1 shows the mean values by gender.

When considering the different age groups, there are significant differences in the mean for some of the variables (Table 4): satisfaction with my place of residence; trust in my neighbours (neighbourhood); my town/city being a safe place to live; satisfaction with my financial situation; satisfaction with my place of work and education; and the effect of the COVID-19 pandemic on my quality of life. For the values obtained with a level of significance below 0.05, the null hypothesis (H0) of equality of means established in the ANOVA is rejected, and we consider that the mean values registered in the different age groups were different.

Considering these results, we conducted a comparative analysis of the different groups to identify the groups between which the differences appeared. To do so, we carried out a post hoc multiple comparison for those variables with significant differences using the Bonferroni criterion. The results are shown in Table A2. Among the noteworthy values in this analysis, the mean for the variable QOL03, on trust in one’s immediate surroundings, is clearly lower in the group of the youngest respondents, undoubtedly linked to the isolation they suffered due to the pandemic. Furthermore, if we look at the impact of the pandemic (QOL13), the difference between the youth group and adults aged 50–59 is also significant. Regarding the financial situation, that of young people is the worst compared to the rest of the ages in the labour market, which emphasizes their economic and social vulnerability including poorer-quality jobs (see QOL11 in Table A3).

To finish the ANOVA, Table A2 shows the means by age group, including all the variables even though we know that for some of them, the differences in the means are not significant. One noteworthy result is that mean values for satisfaction with the place of residence (QOL01) are very low for younger people, unusually so given that when it comes to measures of happiness, they are traditionally the group—along with the elderly—registering the greatest social happiness. As such, this value makes us wonder whether that unhappiness leads to health vulnerability for psychological reasons and what other factors can lead to this vulnerability. Notable among the high values are those registered for the family situation, financial situation, trust in the neighbourhood and employment situation for citizens aged 50 and over; in comparison, the youngest have trouble finding a qualified job, with their work and education situation (QOL12 has lower values than for the rest of the age groups; see Table A2), and with trust in their surroundings after spending long periods in lockdown in recent months (in addition to the values achieved for QOL03, the average among the young people is significantly lower compared with that for the 50–59 group in terms of QOL10 relative to safety; see Table A2).

Nevertheless, the pandemic is the variable that has most affected the over-70 age group (M = 8.53; SD = 1.64), along with the youngest aged between 20 and 29 years old (M = 8.45; SD = 1.87). Therefore, in terms of vulnerability, in mean values, these two age groups are the most affected. That said, the underlying health reasons are certainly different: mainly physical for the elderly and psychological for the young people. If we look at life satisfaction and the factors that can significantly influence it, the group of the youngest respondents presents lower values for the family situation (QOL02), indicating that they are more affected in the personal sphere. Moreover, as seen in the literature review, young Spaniards present very low levels of satisfaction with their work and/or educational environments (QOL12), assigning the same low values as for the leisure and sports facilities on offer in their area (QOL07).

After the ANOVA, we conducted two linear regressions comparing the results relative to two age groups: the youngest, that is, those under 30 (220 respondents in the sample), and the oldest, that is, those aged 30 and over (688 respondents). The endogenous variable was the result of respondents’ a priori and a posteriori evaluations of satisfaction with their quality of life. The objective of this model was to identify the social factors that determine this quality of life and the proportions of the variance they explained, with the unexplained part being understood as due to psychological and personal development issues specific to the individual.
(1)$$\frac{\left(QOL01_{i}+QOL14_{i}\right)}{2}=\alpha +\overset{k}{\underset{s=1}{\sum }}\gamma_{s}QOL_{si}+\varepsilon_{i}$$

The model in Equation (1) estimates the value of the relationship between the *k* key factors (QOL) and social satisfaction with quality of life for the citizens *i*. We thus quantify this relationship or coefficient *γ* for each factor analysed (Table 1). The mathematical relationship follows a linear formulation, as supported by the analyses conducted by López-Ruiz et al. [40] on citizen happiness, according to the equation where *α* is the independent term and *ɛ* is a random variable with zero expectation, uncorrelated and with constant variance. The measures of goodness of fit for the explanation of variance by the *k* factors can be checked through the adjusted coefficient of determination R^2^. The above-referenced paper uses this model to analyse how gender drives different responses in the key factors for Spanish citizens’ satisfaction, with it being the socio-occupational factors that put women at a disadvantage.

The results of the first model, presented in Table 5, reveal that the youngest citizens consider their family situation to be a relevant factor, as well as their neighbourhood, the quality of the environment and the cultural and sports facilities on offer in their town or city. This age group also feels influenced by their employment situation (assessments of their work) and training/education. Thus, the key factor with the largest coefficient and highest significance is the family situation—which is also the variable studied above that shows the greatest deficit for this group—followed by employment/educational situation, immediate surroundings and care for that environment, and the leisure facilities on offer (culture and sports). While all of this makes them happy, it only explains 45% of the total (adjusted R^2^). Young Spaniards in times of the pandemic thus display clear vulnerability given their valuations of life in their place of residence. This is due both to the social factors, which make much less of a difference to them than to the elderly, and the fact that out of preference, they limit themselves to their family and work environment. They are therefore psychologically much more vulnerable; the pandemic has hit them harder as individuals, opening up an age gap in quality of life driven by social issues.

On the other hand, the results from the second model for adults and the elderly in Spain (Table 6) show that apart from the family (QOL02), which is also the most important factor in their quality of life, they also align on the importance of working conditions and education (QOL12) as the second element, followed by leisure in relation to cultural and sports facilities (QOL04) and the quality of the environment (QOL04). In this case, key factors also include issues of mobility and access to consumption (QOL05), as well as the safety of the entire surrounding area, not just their neighbourhood (QOL10). The fundamental difference is that these social conditions account for 62% of quality of life for this group, with a drastically reduced emphasis on the individual issues and also the psychological issues for these citizens.

Finally, we analysed whether the results differed significantly by age and gender regarding the question of which sphere of citizens’ lives has been most affected by the pandemic. This is not included in Table 1 and refers to a question from Section 2.1, where we ask about which aspects of their lives had been the most affected by the pandemic. Specifically, this last section focuses on the respondents’ surroundings, distinguishing aspects such as personal and family situation; employment and economic situation; training and education; and place of residence; or not having been affected by the pandemic.

To do so, we used a chi-square test to determine whether gender and/or age group has a relationship of dependence with the most affected sphere. The results show chi-square values of 13.590 when considering the relationship with gender and 83.753 with age group, and in both cases the relationship between the factors is significant at 5%.

Table 7 and Table 8 explain the percentage distribution of the sample, accounting for the areas they consider most affected by gender and by age group, respectively. Women are more affected in family and socio-occupational terms than men, so the vulnerability profile is biased in this aspect since these two areas both explain citizens’ quality of life.

In terms of age, there are two particularly noteworthy findings: first, that young people are those who least feel that they have not been affected by the pandemic (0% in the case of the youngest and 3.5% for those aged 20–29), and second, that the work environment and education are the aspects clearly identified as the most sensitive issues for this group, with large differences relative to the other groups.

## 4. Discussion

The citizens whose quality of life suffered the most with the emergence of the pandemic were young people aged between 18 and 29 years old and those aged over 70. The findings of this research reveal that social factors are relevant for only 45% in the young people group, while for the oldest group, this figure exceeds 62%. This finding is in line with the studies by Anderson and Gettings [8] and Rahman and Jahan [18], who concluded that despite the fact the elderly have coped better with the pandemic in psychological terms, they have become an at-risk population, placing them among the most vulnerable citizens.

The ANOVA revealed that gender did not drive differences in the means of the different variables that affect quality of life in this study. However, differences did appear with age, for example, in satisfaction with the place of residence, the neighbourhood, the perceived safety of living in that place, the financial and employment situation, and of course, the perception of the effect of the pandemic. Regarding trust in one’s immediate surroundings (QOL03), of all the established age groups, it was the youngest who registered the lowest values. Differences in the perceived impact of the pandemic were also shown with respect to the group aged between 50 and 59 years old (QOL13). The youngest also showed dissatisfaction with their work and/or educational environment (QOL12) and pointed to inadequate leisure and sports facilities in their area (QOL07). This coincides with findings reported in previous literature, such as by King et al. [37] or Almeida and Santos [20].

Therefore, in terms of gender and age, the data show, on the one hand, how the gap in this question widens in relation to women’s employment situation and family issues. In this case, the vulnerability profile is biased in regard to these aspects since they are the ones that explain citizens’ quality of life. This is consistent with studies such as that by King et al. [37] but in other aspects aligns with that by Bau et al. [38], who concluded that gender differences which pre-date the pandemic are being exacerbated. On the other hand, in terms of age, the results reveal that young people are the most affected in terms of the work and educational environment.

Furthermore, regression models showed that young Spaniards in times of pandemic present a vulnerability due to social factors in their place of residence, such as family and work, which make much less of a difference to them than to the elderly. As for the similarities revealed by the two linear regression models, they point to improvement measures that can be deployed among the youngest age group. This is in line with the study by Anderson and Gettings [8].

Finally, the quality-of-life factors found to be the most affected by COVID-19 were the family environment, and to a lesser extent the work environment, followed by the educational sphere. Only 6% of the population answered that the pandemic had not affected them at all.

## 5. Conclusions

Psychological vulnerability has been exacerbated by COVID-19, particularly for young people, specifically those under 30 years of age. Some of them suffered major disruptions to their education due to the closure of classrooms, while others saw their financial emancipation hindered by the difficulty of entering the labour market. In addition to this situation, the isolation suffered has meant that the youngest display a lack of trust in their immediate surroundings. Furthermore, taking into consideration gender, young women suffered more from this situation than men, being unable to balance their work and family lives due to having to care for children or other family members during the lockdown. It is also proven that there is a gender gap in terms of women’s self-assessment of their employment. Finally, to aggravate this social mix, the social factors that determine quality of life are more representative of the elderly, with young people being more affected by their own individual conditions, such as psychological aspects. Psychological factors such as self-esteem are difficult to measure exactly, but the results do show how these factors have been more decisive in the quality of life of young people than have physical factors in relation to the elderly. This is in line with other works such as those by Anderson and Gettings [8] and Rahman and Jahan [18].

On the other hand, this study indicates that vulnerability also falls on the older Spanish population (found to be worse among people aged over 70), affecting their physical condition, due to the isolation suffered during the pandemic and to health and immunity issues.

Notable among the factors that play a relevant role in this vulnerability for over 30s are the family situation (far more so for those over 70), their employment and education conditions and the cultural and sports facilities on offer, followed by the quality of the environment. Key factors also include issues of mobility and access to consumption, as well as the safety of their entire surrounding area, not just their neighbourhood.

Therefore, it can be concluded from this research that of the sample of Spanish citizens compiled, it is those over 70s who have the most problems with vulnerability, along with the youngest citizens aged between 18 and 29 years old. However, in this case, the analysis carried out has not confirmed an impact of gender except in regard to family and work situations, where women are more affected. This can indirectly aggravate the situation they face due to the pandemic for any age group.

Lastly, this research presents some limitations that are being addressed with future studies. The first refers to enlarging the sample of citizens. Another was that it would have been interesting to study other sociodemographic variables of Spanish citizens that made them more vulnerable depending on the employment situation in certain environments. In addition, in future lines of research, it would be interesting to examine whether, after the pandemic, citizens consider these variables relevant or highlight others that were not considered at the time.

## Figures and Tables

**Table 1 ijerph-19-10269-t001:** Variables.

Variable	Code
In general, I feel good, and I am very satisfied with life in my place of residence.	QOL01
My family situation is satisfactory.	QOL02
In general, I can trust the people in my neighbourhood.	QOL03
In my town/city, the air quality, noise and light pollution, and cleanliness are acceptable.	QOL04
Public transport, access to and availability of shops and shopping areas are adequate.	QOL05
In my town/city, there are enough green spaces/adequate green spaces.	QOL06
The cultural and sporting facilities on offer, as well as the possibility of performing sporting activities, are adequate.	QOL07
Health services—hospitals, health centres and health care workers—are adequate and accessible from my place of residence.	QOL08
The educational offer at all levels, including university, is adequate and accessible from my place of residence.	QOL09
My town/city is a safe place to live.	QOL10
Overall, I am satisfied with my financial situation.	QOL11
In the work I do or have done, I feel fulfilled and valued. In short, I am satisfied with my employment situation or, where applicable, educational situation.	QOL12
The COVID-19 pandemic is affecting my quality of life.	QOL13
Having analysed all the dimensions, indicate your degree of happiness on this scale.	QOL14

**Table 2 ijerph-19-10269-t002:** Sample characteristics.

Variable	Categories	Percentage (%)
Civil Status	Single	27.1
	Married or in a stable relationship	65.1
	Divorced	6.8
	Widowed	1.0
Gender	Male	38.8
	Female	61.2
Age	18 or 19 years old	2.0
	20–29 years old	22.2
	30–39 years old	12.0
	40–49 years old	30.7
	50–59 years old	23.3
	60–69 years old	8.0
	Over 70 years old	1.7
Economic sector	Business	5.6
	Construction	2.1
	Education	25.8
	Unemployed or inactive	21.8
	Energy, water	1.8
	Finance and insurance	6.6
	Manufacture	1.1
	Other services (communication, information …)	21.5
	Agriculture, animal husbandry, fishing (Primary)	1.8
	Health care	7.6
	Transport and storage	2.0
	Tourism, catering, hospitality	2.4
Place of residence	≤1000-inhabitant municipality	5.6
	Municipality of 1001–5000 inhabitants	2.1
	Municipality of 5001–100,000 inhabitants	25.8
	>100,000-inhabitant municipality	21.8

**Table 3 ijerph-19-10269-t003:** Levene’s test for variance homogeneity.

Variable	Gender		Age	
	Statistic	Sig.	Statistic	Sig.
QOL01	1.396	0.238	2.834	0.010
QOL02	0.627	0.429	1.828	0.091
QOL03	0.881	0.348	4.200	0.000
QOL04	0.044	0.834	0.791	0.577
QOL05	3.040	0.082	1.093	0.365
QOL06	5.216	0.023	1.687	0.121
QOL07	7.319	0.007	0.991	0.430
QOL08	0.240	0.624	1.329	0.241
QOL09	4.620	0.032	5.020	0.000
QOL10	0.034	0.853	0.151	0.989
QOL11	1.215	0.271	3.762	0.001
QOL12	7.336	0.007	5.056	0.000
QOL13	0.296	0.586	1.536	0.163
QOL14	0.524	0.469	0.181	0.982

**Table 4 ijerph-19-10269-t004:** ANOVA analysis: Gender and Age—quality of life variables.

Variable	Gender		Age	
	Statistic	Sig.	Statistic	Sig.
QOL01	1.327 (F)	0.250	3.250 (F)	**0.004**
QOL02	0.869 (F)	0.351	1.185 (F)	0.312
QOL03	0.540 (F)	0.463	5.734 (F)	**0.000**
QOL04	1.618 (F)	0.204	0.627 (F)	0.709
QOL05	4.407 (F)	**0.036**	1.619 (F)	0.139
QOL06	1.073 (W)	0.301	1.090 (W)	0.373
QOL07	1.119 (W)	0.290	0.421 (W)	0.863
QOL08	0.532 (F)	0.466	0.772 (F)	0.592
QOL09	0.949 (W)	0.330	1.673 (W)	0.135
QOL10	0.063 (F)	0.802	2.672 (F)	**0.014**
QOL11	0.456 (F)	0.500	3.897 (F)	**0.001**
QOL12	2.452 (W)	0.118	11.587 (W)	**0.000**
QOL13	0.280 (F)	0.597	2.590 (F)	**0.017**
QOL14	0.169 (F)	0.681	1.337 (F)	0.238

Note: (W) Welch statistic; (F) F statistic. Data in bold highlight existence at the 5% level.

**Table 5 ijerph-19-10269-t005:** Vulnerability in the youngest group (age < 30 years).

Variable/Factor	Coefficient	*t*-Statistic
Independent	1.956660	4.603004 ***
QOL02	0.299226	6.818505 ***
QOL03	0.108798	2.677405 ***
QOL04	0.088226	2.503731 **
QOL07	0.139656	3.725294 ***
QOL12	0.109675	3.153636 ***
R^2^	0.466	
R^2 ADJUSTED^	0.453	
N	220	

Note: significance level ** *p* < 0.05; *** *p* < 0.01.

**Table 6 ijerph-19-10269-t006:** Vulnerability in the older group (age ≥ 30 years).

Variable/Factor	Coefficient	*t*-Statistic
Independent	1.040443	4.747777 ***
QOL02	0.407903	20.87580 ***
QOL04	0.047278	2.533237 **
QOL05	0.089284	4.653127 ***
QOL07	0.096980	4.569368 ***
QOL10	0.054199	2.058057 **
QOL12	0.163941	9.038410 ***
R^2^	0.627	
R^2 ADJUSTED^	0.624	
N	688	

Note: significance level ** *p* < 0.05; *** *p* < 0.01.

**Table 7 ijerph-19-10269-t007:** Most affected spheres of citizens’ lives by gender.

		COVID-19 Has Affected My Life More In				
		Personal and Family	Work and Economic	It Hasn’t Affected Me	Education and Formation	Home
Gender	Male	58.8	17.9	8.8	11.1	3.4
	Female	65.6	18.7	4.1	10.1	1.4
Total		63.0	18.4	5.9	10.5	2.2

**Table 8 ijerph-19-10269-t008:** Most affected spheres of citizens’ lives by age group.

		COVID-19 Has Affected My Life More In				
		Personal and Family	Work and Economic	It Hasn’t Affected Me	Education and Formation	Home
Age	18–19	55.6	5.6		38.9	
	20–29	54.5	17.3	3.5	22.8	2.0
	30–39	68.8	19.3	8.3	2.8	0.9
	40–49	64.2	19.7	5.4	7.9	2.9
	50–59	64.6	21.2	7.5	5.7	0.9
	60–69	67.1	11.0	9.6	6.8	5.5
	Over 70	80.0	13.3			6.7
Total		63.0	18.4	5.9	10.5	2.2

## Data Availability

The data supporting the reported results can be found at https://blog.uclm.es/victorlopez/ (accessed on 15 May 2022).

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
