# Peer review of "Psychological and Social Vulnerability in Spaniards’ Quality of Life in the Face of COVID-19: Age and Gender Results"

_ijerph, 2022, doi:10.3390/ijerph191610269_

Round 1
Reviewer 1 Report
General comments
Very interesting theme, and of high relevance to research.
I present my reflections according to your article.
Title
“Psychological vulnerability in young Spaniards' quality of life in the face of COVID-19” - I don't think it translates the work presented.
. It samples young Spaniard’s, but in the sample only 24.2% are young (18 and 29 years according to the definition they present in the introduction) and it may still be questionable whether this period of the life cycle is still designated as young people (according to other sources- UN and Pieh et al.).
. 908 Spanish citizens – and of those only about 218 are young. Can you talk about young Spaniards?
Abstract
. It would be interesting to present/identify the sample and evaluation instruments that were used in the study
. You talk about more vulnerable groups. It's not understandable what vulnerability you're referring to. If to the virus, if to the period of life-cycle- young/old, if to the place of residence or none of those. I consider it important to contextualize. It will also be important to frame the definition of vulnerability.
. You approach age as a study variable, however, in the title, you put young people but then approach age (periods of life) as a variable of study and talk about vulnerable groups. It's important to clarify.
. The results presented here are unclear, they do not determine the directionality of the phenomenon under study.
Introduction
. Lines 49 – 72- You are not specifically evaluating this period of life, it does not seem necessary to me this explanation so exhaustive and explanatory in your article.
. Review crow at line 95 “education In this regard, other researchers such as Perry and McConney…”
. Review crow at line 123 “Lastly,, Silva Moreira et al.”
. On line 138, the following sentence is not framed “Young people who suffer from a chronic illness or who are close to someone who is ill tend to suffer psychological consequences (20)”.
. You define as objective “the aim of this research is to analyse the most vulnerable groups established by gender and by age range, accounting for the influence of COVID-19 on certain factors influencing Spanish citizens' quality of life; namely, living conditions, family situation, employment status, income level and well-being.” And the title of the article is “Psychological vulnerability in young Spaniards' quality of life in the face of COVID-19”- has to be, in my view, reworked- the title or objective - in the title you speak of psychological variables, in the objective you introduce sociodemographic characteristics. The objective is not clearly formulated.
. I suggest reflection and contextualization of the issue “the most vulnerable groups established by gender and by age range”. It's not clear.
. You refer to that “we conduct a review of vulnerability in citizens based on the determinants of their quality of life, and how the pandemic has affected them depending on the sociodemographic characteristics of age and gender.” How?
. You refer to that “By conducting an analysis of variance (ANOVA) and a regression analysis, we establish two age groups”- Two groups in relation to the different age groups in the analysis of variance? Wasn't it?“ To this end, different age groups were taken into account: 18 and 19 years old; between 20 and 29 years old; 30 to 39 years old; 40 to 49 years old; 50 to 59 years old; 60 to 69 years old; and over 70.”? And for the regression analysis “two age groups were established in order to be able to fit two models and compare the results: young people, according to the INJUVE criterion”? Even so, you talk about a sample distribution of 24.2% (18-29 years old) and 75,7% (30 and over 70 years old).
Materials and Methods
. You refer “socio- demographic data on Spanish citizens, such as age, gender, marital status, place of residence, and job sector”. “In this case, age and gender are the characteristics we will use to differentiate between respondents' vulnerability.” Which means that you have assessed psychological vulnerability based on these characteristics and indicators?
. You refer “The second section captured variables related to the measurement of citizens' quality of life and happiness.“ Happiness appears here for the first time. It should be previously framed.
. You refer “key aspects of quality of life and the respondents' surroundings, such as family situation, trust in the neighbourhood, trust in the foreign-born population, care for the environment, commercial accessibility and public transport, cultural and sports facilities and activities in their place of residence, available health and educational services, housing prices and safety of the residential environment” – you're talking about social vulnerability?
. You refer “On the other hand, it included items related to respondents' educational and employment situation; that is, employment status, education received, working environment, remote work, and financial situation”- once again you are talking about characteristics that can lead to social vulnerability?
. In “Table 1. Variables” are variables or questions/items related to key aspects of quality of life? I suggest you put this information in support material.
. You refer “The mean score for this answer also provides the first indications on this question, with the negative influence on quality of life reaching 7.96 points.” But previously you had given the following information “answered on a 1-10 Likert-type scale, where 1 indicated “not at all satisfied” and 10 “very satisfied”. How do you describe negative influence on quality of life? Based on what parameters? It is not clear how you evaluate.
. Statistical values should be presented in accordance with the standards of the APA “However, the results showed that people aged between 20 and 29 had a higher mean score of 8.45, with those over 70 registering an even higher mean of 8.53.” That is, “However, the results showed that people aged between 20 and 29 had a higher mean score (M=8.45; SD=), with those over 70 registering an even higher mean (M=8.53; SD=).”
Results
. In “Table 4. ANOVA analysis: Gender and Age – quality of life variables” you didn't put the sub-title. What does W and F mean? Although you have described it before, I suggest you describe it in the sub-title W- Welch statistic; F- F statistic.
. In QOL05, M=.036, how do you conclude “Therefore, it can be concluded that there is no difference in the mean values assigned to the different variables that affect quality of life or in their evaluation when comparing the groups of men and women.”? Legitimizing with a distinct item (QOL05, M=0.118)? It would be important to explain this issue better. I don't think it's pertinent.
. Lines 273-279, what direction? Not displayed.
. In line 278 you talk about a null hypothesis that until then had never been addressed “the null hypothesis (H0) of equality of means is rejected”. If you want to keep this, you must present first the hypotheses.
. The results should be presented in accordance with the standards of the APA (7Ed). So they're very confused.
Discussion
. Very generalist- “Based on the results of this study, we identify profiles of vulnerability to the pandemic for Spanish citizens, according to their age and gender, and taking into account aspects of quality of life. The factors that determine this quality of life have been affected by the emergence of the pandemic, more specifically in certain age groups, and particularly in terms of employment, educational and family situations.” This paragraph does not bring effective knowledge.
. You refer “Psychological vulnerability has been exacerbated by COVID-19 particularly for young people, specifically those under 30 years of age. Some of them suffered major disruptions to their education due to the closure of classrooms, while others saw their financial emancipation further hindered by the difficulty of entering the labour market.” I question again, psychological vulnerability can be assessed through the closure of classrooms, financial emancipation and the difficulty of entering the labour market?
. You refer “young people being more affected by their own individual conditions, such as psychological aspects.” How did you evaluate these psychological aspects? Self-esteem? Self-concept? Perception of psychoemotional and relational adjustment? It is not clear how you evaluated these aspects? It is important to explain.
. You say that the research presents some limitations- however you only identify the enlarging of the sample of citizens. There are no other limitations?
Reviewer 2 Report
Review of “Psychological vulnerability in young Spaniards’ quality of life in the face of COVID-19”
20 July 2022
This manuscript describes a study looking at factors of quality of life and age for a sample of Spaniards during COVID.
Overall, the work has merit, but the manuscript needs a significant amount of revision before it is ready to be published. I have only a few comments on the study itself; most of my concerns are in the discussion of results.
Line 172: is this the average margin of error across the whole survey?
Line 195: For the questions as translated, “satisfied” or not “satisfied” do not seem to be the appropriate word. Probably a translation issue. Maybe “applicable” or “true” or “agree”?
Table 1: I was not clear on how QOL13 and QOL14 related to the data in scale 7. This should be clarified.
Line 213: I think you mean young people (30 and younger) and less young (over 30). If so, please add this for clarity.
Lines 233-239: It would be helpful to add a reminder here that scores on an individual question range from 1 to 10, so 7.96 is put in perspective.
Table 3 and Table 4 do not seem to match up in terms of using the W or F statistic. Please recheck this data.
I also wanted to see the mean scores for gender; adding a table in the appendix would be enough. This is so the reader can make sense of not only gender differences but also overall responses.
As mentioned, my biggest concerns are in the discussion of results. I find that there are many statements which do not match the data or are overstated from the results shown. I will give a few examples of these.
Lines 265-268: I don’t see any connection between QOL5 and QOL12, and since QOL12 is only at p=.118, it should not be included unless there is something interested in the actual values, not the gender difference.
Lines 280-288: I see that QOL3 is very interesting and should be mentioned, but I don’t see why the authors note only QOL13 when there are many other significant results.
Lines 298-300: This statement does not come from the data; if it comes from the data plus extant research, it should be mentioned.
Often QOL01 is mentioned as satisfaction with place of residence, but the question is broader than that. It reads as general life satisfaction, not just happiness with location. The phrasing and discussion regarding this factor needs to be re-thought.
Lines 363-364: As mentioned above, I don’t understand where this data came from. QOL14 or something else?
The conclusions need to be clear about what information came from the literature and what came from this research. Otherwise there are statements that do not have evidence for them and/or are overstated.
Please note that these are simply examples of my concerns with the discussion and conclusion sections. The authors should go through line by line in these sections to make sure they are not overstating or hypothesizing or drawing from literature instead of this study.
Reviewer 3 Report
Overall, this study investigates an important topic during the COVID-19 pandemic. However, there are several methodological concerns, I raise them here:
1. Research design
Can you clarify limitations of the convenient sample approach using online survey distributed by email and social media? How does your sample compare to the typical target population? (i.e. age group, demographics, income-level, etc.)
2. Construct measures
Can you provide additional psychometric properties indicating reliability and validity of QOL instruments?
3. Analysis and results
There are many p-values close to 0.05 (in Tables 3-4) and several t-statistic close to 1.96 (in Tables 5-6). Considering that you are conducting multiple hypotheses testing in the same sample, the probability that the significant result happens just due to chance is increasing exponentially with the number of hypotheses tested. How did you address/account for this design feature?
4. Discussion
The existing discussion is quite weak. Implications of the present study should be elaborated. It would be useful to consider and add the latest literature the following studies that cover the topic of COVID-19 and psychosocial stress, particularly its related causes and consequences of COVID-related health outcomes in young populations. I randomly searched a couple key words relevant to your study on Web of Science and think you should consider citing these relevant studies:
Reviewer 4 Report
Dear Authors,
The method you use and the analyses of your work is appropriate however I would suggest the following issues:
1- It is better to add a new session, that of the literature review to clearly identify the literature gap and the importance of this study. You have add some literature in the introduction part but that is not sufficient.
2- Reference are limited and I would recomment to add som more that are similar studies with yours. This would help to clarify the questions raised in you paper.
I hope these notes would contribute to the enhancement of your paper.
Best regards,
Round 2
Reviewer 1 Report
see attachment

Reviewer 2 Report
Thank you for making changes. I am still unclear on where the data in tables 7 and 8 came from in terms of the 14 QOL questions. Was there another question asking people to rank the area of highest effect?
Reviewer 3 Report
The revision is acceptable, but the authors might still need some language editing to weed out grammatical issues which hinder the flow of the paper.
Reviewer 4 Report
The recommendations have been taken into consideration, thus I recommend its publication.
